# Intelligent Biological Networks: Improving Anti-Microbial Resistance Resilience through Nutritional Interventions to Understand Protozoal Gut Infections

**DOI:** 10.3390/microorganisms11071800

**Published:** 2023-07-13

**Authors:** Avinash V. Karpe, David J. Beale, Cuong D. Tran

**Affiliations:** 1Agriculture and Food, Commonwealth Scientific and Industrial Research Organisation, Black Mountain Science and Innovation Park, Acton, ACT 2601, Australia; 2Socio-Eternal Thinking for Unity (SETU), Melbourne, VIC 3805, Australia; 3Environment, Commonwealth Scientific and Industrial Research Organisation, Ecosciences Precinct, Dutton Park, QLD 4102, Australia; david.beale@csiro.au; 4Health and Biosecurity, Commonwealth Scientific and Industrial Research Organisation, Gate 13 Kintore Ave., Adelaide, SA 5000, Australia; cuong.tran@csiro.au

**Keywords:** systems biology, multiomics, *Cryptosporidium*, *Giardia*, *Entamoeba*, anti-microbial resistance, multi-drug resistance, probiotic, prebiotics, synbiotics, postbiotics

## Abstract

Enteric protozoan pathogenic infections significantly contribute to the global burden of gastrointestinal illnesses. Their occurrence is considerable within remote and indigenous communities and regions due to reduced access to clean water and adequate sanitation. The robustness of these pathogens leads to a requirement of harsh treatment methods, such as medicinal drugs or antibiotics. However, in addition to protozoal infection itself, these treatments impact the gut microbiome and create dysbiosis. This often leads to opportunistic pathogen invasion, anti-microbial resistance, or functional gastrointestinal disorders, such as irritable bowel syndrome. Moreover, these impacts do not remain confined to the gut and are reflected across the gut–brain, gut–liver, and gut–lung axes, among others. Therefore, apart from medicinal treatment, nutritional supplementation is also a key aspect of providing recovery from this dysbiosis. Future proteins, prebiotics, probiotics, synbiotics, and food formulations offer a good solution to remedy this dysbiosis. Furthermore, nutritional supplementation also helps to build resilience against opportunistic pathogens and potential future infections and disorders that may arise due to the dysbiosis. Systems biology techniques have shown to be highly effective tools to understand the biochemistry of these processes. Systems biology techniques characterize the fundamental host–pathogen interaction biochemical pathways at various infection and recovery stages. This same mechanism also allows the impact of the abovementioned treatment methods of gut microbiome remediation to be tracked. This manuscript discusses system biology approaches, analytical techniques, and interaction and association networks, to understand (1) infection mechanisms and current global status; (2) cross-organ impacts of dysbiosis, particularly within the gut–liver and gut–lung axes; and (3) nutritional interventions. This study highlights the impact of anti-microbial resistance and multi-drug resistance from the perspective of protozoal infections. It also highlights the role of nutritional interventions to add resilience against the chronic problems caused by these phenomena.

## 1. Introduction

One of the biggest concerns in human health is enteric infection. Besides causing considerable malnutrition, they also contribute to the death of a significant number of people worldwide. In spite of the supplementation programs of the World Health Organization (WHO) in recent decades, 1.2 million annual deaths were reported in 2006 [1]. Globally, 1.4 million deaths were attributed to intestinal infectious diseases in 2015, including 0.53 million deaths among children under 4 years of age [2] and in developing/underdeveloped countries (Figure 1).

Protozoan infections contribute towards a large proportion of global mortality, causing up to 0.84 million deaths annually, as recent as 2015 [3]. Detailed information related to the spread of these organisms is provided in Table 1. While the widespread outbreak of *Entamoeba* spp., particularly *E. histolytica*, is less highlighted, it causes the second most common parasite-based human deaths worldwide (after malaria). About 10% of the human population is infected by *E. histolytica*, resulting in 110,000 annual deaths [4].

The prevalence of *Giardia* spp. is much higher in developing and underdeveloped countries. Children under 10 years of age accounted for up to 20% of global outbreaks [5] and 35.2% of global waterborne disease outbreaks between 2004 and 2010.

The number of deaths caused by *Cryptosporidium* spp. has been estimated at 83,000 (2005), 99,800 (2010), and 64,800 (2015). In 2016, there were approximately 2.7 million reported cases of pediatric cryptosporidiosis in South and Latin America, 3.5 million in Sub-Saharan Africa, and 3.2 million in Asia, with 4.7 million cases in the Indian subcontinent. There is a high prevalence of entamoebiasis and cryptosporidiosis in Australia, especially during summer. There were about 18,000 reported cases in 2010, most of them in Aboriginal and Torres Strait Islander people [6]. Out of all the clinical cases from 88 countries between 2000 and 2015, about 10.9% of the population was infected with cryptosporidiosis [6]. It should be noted that outbreaks of intestinal waterborne protozoan are mostly observed in developed countries and is less common in developing/underdeveloped ones. Also, although related to childhood diarrhea, in endemic countries, these infections are usually asymptomatic and only are diagnosed during coprological surveys of intestinal parasites among school children.

This short review discusses protozoal gut infection from three perspectives. Firstly, it provides an overview of the impacts of these infections on the global population. Secondly, it highlights the gut microbiome and ecosystem perturbations these infections cause, and extra-gut impacts of these perturbations. Lastly, it discusses the potential nutritional advances of the replacement or adjuvants to contemporary treatment drugs that can be utilized to minimize or eliminate the infections without developing multi-drug resistance and anti-microbial resistance. In this review, these perspectives are discussed using multiomics platforms that have been rapidly advancing over the last decade. Although the review does not include a detailed elaboration of infection mechanisms and technical aspects of omics platforms, it provides the sources that readers can refer to for further information.

**Figure 1 microorganisms-11-01800-f001:**
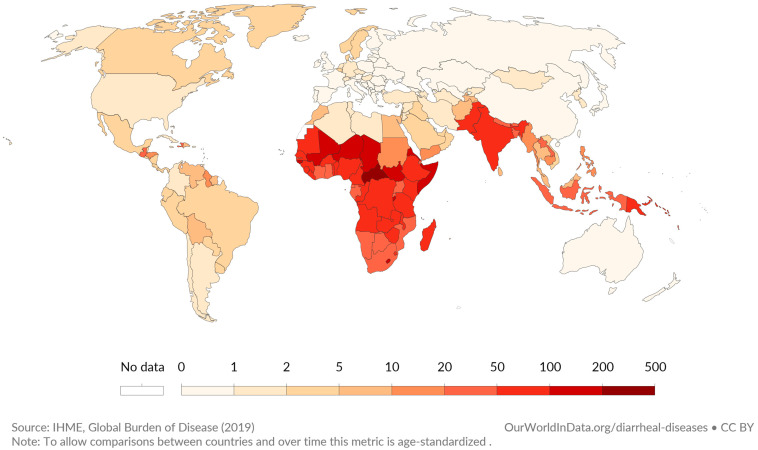
Global distribution of estimated deaths caused by diarrheal infections per 100,000 people in 2019 [7]. The interactive map can be obtained from https://ourworldindata.org/grapher/diarrheal-disease-death-rates (accessed on 28 June 2023). All visualizations, data, and code produced by “Our World in Data” are completely open access under the Creative Commons BY license.

**Table 1 microorganisms-11-01800-t001:** Transmission of key protozoal enteric parasites and their current global status (2000–2016).

Name	Transmission Mode	Health Symptoms	Hosts	Disinfection Resistance	Outbreaks/Cases	References
*Cryptosporidium* spp.	Water (drinking and recreational), fecal-oral route	Moderate/diarrhea. Mostly asymptomatic, diagnosed during clinical presentation	Humans, cattle, rodents, wild animals	Very high/ozone (4 ppm/10 min); >3% hypochlorite	239/65,540 (2004–2014)	[2,3,6,8]
*Giardia* spp.		Moderate/diarrhea, gas, bloating, anorexia. Mostly asymptomatic, diagnosed during clinical presentation	Human, cattle, wild animals	High/fenbendazole (5 mg/kg); ozone (0.3 ppm/3 min); 1% sodium hypochlorite	142/1110 (2007–2014)	[1,3,8,9]
*Entamoeba histolytica*		Severe/colitis, dysentery, diarrhea, liver abscess	Humans, non-human primates	High/chlorine (5 ppm, pH 7, 5 min), 1% sodium hypochlorite	15/9.41 million (2000–2015)	[2,4,9,10]

## 2. Research Methodology

Due to the wide horizon of this review, a variety of keywords were used to search the development in research and development phases over the last 15 years. The keywords include enteric infection, Protozoa, *Cryptosporidium* (particularly, *C. parvum*), *Giardia* (particularly, *G. duodenalis*), *Entamoeba* (particularly, *E. histolytica*), diarrhea, multi-drug resistance (MDR), anti-microbial resistance (AMR), host–parasite–microbiome relationships, extra-gut effects, inflammation, malabsorption, immune system, prebiotics, probiotics, synbiotics, anti-microbial proteins, pathogen/infection pathways, metagenomics, transcriptomics, metabolomics, and proteomics. The technical aspects of multiomics platforms, such as bioinformatics tools and toolboxes, statistical analysis methods, and research protocols, are beyond the scope of this review and are excluded.

This review primarily covers the experimental findings of clinical and animal trials, in vivo and in vitro experiments, case studies, and annual reports. We did not conduct a systematic review of reviews to prevent introducing too much statistical power and the meta-analysis effect. However, some key reviews are presented, which readers can refer to for a deeper understanding of the aspects of individual perspectives that this review covers.

## 3. Life Cycle and Infection Mechanism

### 3.1. Life Cycle

*Cryptosporidium* spp., *Giardia* spp., and *Entamoeba* spp. Spread through aqueous routes, either via drinking water, food, fecal matter, or environmental calamities associated with water. Cysts are often the first step of infection by these protozoan parasites, originating from infected matrices or individuals. The infection occurs by the ingestion of tetranucleated cysts (*Entamoeba histolytica* and *Giardia*) or sporulated oocysts (*Cryptosporidium*). Acidic conditions in the stomach, followed by slightly alkaline conditions in the upper intestine (duodenum), result in the process of excystation, releasing these trophozoites or sporozoites into the duodenum [11,12].

The trophozoites attached to the intestinal mucosa are mostly localized and non-invasive. But sometimes, particularly with *Entamoeba*, trophozoites do penetrate the mucosal and epidermal layers and are carried into the bloodstream to cause multi-organ infections [11,13]. *Entamoeba* trophozoites are also particularly motile because of their pseudopodia, giving them the ability to move actively towards the larger intestine. Trophozoites then reproduce asexually by binary fission in the large intestine and feed on cellular debris and microbiota.

*Giardia*, on the other hand, reproduce by asexual binary fusion and then attach to the apical epithelium layer of the duodenal and jejunal parts of the small intestine [12].

*Cryptosporidium* reproduces through both sexual (gemetogeny) and asexual (merogeny) reproduction in the epithelial cells of the intestine. Immature oocysts sporulate inside the host and are released as mature cysts in feces (Figure 2).

*Cryptosporidium* and *Giardia* are stable interaction parasites that cause asymptomatic infections. Therefore, they do not cause significant changes in host physiology within their preferred habitat of the small intestine, resulting in a delayed diagnosis of infection. However, the localized nature of pathogens also mean lower mortality, except for the immune-compromised host [14]. In the case of *Entamoeba*, since the parasite can penetrate the epithelial lining of the intestines and enter the bloodstream, the mortality rate exceeds 50% [13].

### 3.2. Dysbiosis and Target Organs

Although most enteric protozoans are limited to intestinal systems, there are numerous variations by which they affect the host. These variations include nutrient absorption and host microbiome perturbation, which result in impacts on other organs. This may also affect biochemical and physiological conditions in the other organs and systems of the host body (Figure 3).

#### Dysbiosis

Bacterial species in the large intestine occur as micro-colonies or associative multispecies consortia. Studies of oral biofilms and assemblages in the large intestine have shown that the bacteria reside as biofilm communities in the cavities along the mucosal lining on the epithelial lining in both clinical [15] and animal [16] models. It is well known that *Cryptosporidium* spp. is an epithelial-lining-based parasite, causing minimal invasion; generally, it is unable to penetrate through the mucosal layers. Therefore, it is highly likely that its interactions with bacterial biofilms in the gut lining cause behavioral changes in itself and the surrounding bacterial community. Recently, Koh, et al. [17] demonstrated a direct proportionality of *Cryptosporidium* growth with respect to Pseudomonas aeruginosa biofilm maturation, showing a 2–3-fold increase in a population of *Cryptosporidium* under an aquatic environment. Also, genomic studies on Coquerel’s sifaka, a Madagascar lemur species, indicated considerable microbial diversity depletion with recovery depending on the host age (older lemurs recovered earlier than younger ones). The number of bacteria associated with human enteric diseases, such as *Desulfovibrio* spp. and Enterococcus spp., increased considerably. On the other hand, the population of healthy gut bacteria, such as *Bifidobacterium* spp., *Akkermansia* spp., and *Succinovibrio* spp., decreased considerably in infected individuals [18]. It has been reported that a release of several enzymes, such as lactate dehydrogenase, proteases, hemolyses, and phospholipases, is caused during *C. parvum* infection. Resulting cellular hydrolysis and degradation causes blunting of microvilli and measurable intestinal epithelium damage, likely affecting the microbiome (rather than *C. parvum* directly affecting microbiome populations, as seen in *Giardiasis*) [19]. Our multiomics study of mouse cryptosporidiosis indicated that during the infection, the *Faecalibaculum* and *Lachnospiraceae* populations depleted from the duodenum onwards. However, the *Lactobacillus*, *Lachnospiraceae*, *Desulphovibrio*, and *Coriobacteria* populations were elevated in the jejunum and ileum [20].

The endosymbiont interaction of parasite and bacterial species has been shown in a studies performed with mouse models of *Giardia* infection [21]. It was shown that the inherent gut microbiota interferes with *Giardia* infection. However, in the presence of the bacterial endosymbionts, such as *Escherichia coli* and *Shigella* spp., which are responsible for enteropathogenicity, the intestinal saccharide ligands change, aiding the protozoan parasites in colonizing the sites by adhesion [22]. Briefly, the study indicated that the bacterial symbiotes altered the cellular surface saccharides of *Crithidia* oncopelti, a protozoal parasite, through the fucose-binding lecithin, increasing agglutination. Besides protecting trophozoites, this activity also increases *G. duodenalis* parasite expression [23]. However, a similar interaction under in vitro condition has shown to increase the pathogen virulence (in addition to trophozoite protection and increasing expression) of *Entamoeba* spp. Infection [24,25].

A more recent study on *Giardia*’s effect on microbiota was reported by Caenorhabditis elegans model analysis through genomic output [26]. The study involved a multi-pronged approach, where interactions between G. lamblia and commensal *E. coli* were tested. Additionally, the microbiota from healthy and irritable bowel syndrome (IBS)-affected human representatives were transplanted in C. elegans intestine, and the effects of *Giardia*, *E. coli*, and ‘*Giardia* + *E. coli*’ were observed. It was seen that *Giardia* altered *E. coli* gene expressions, especially of ribosomal proteins, flagellar, adhesion, and transport (taurine) genes. This resulted in an increased virulence in commensal *E. coli*, converting them from host–microbiome symbiotic species into pathogenic species. The interaction also decreased the expression of cysB genes, responsible for producing H_2_S [26], which is known to be an anti-inflammatory and cryoprotective metabolite in the intestine [27]. We have also noted that during *Giardiasis* infection in mice, the populations of *Autopobiaceae* and *Desulfovibrionaceae* increased, while that of *Akkermansiaceae* decreased [28], in the gut.

Similar observations have also been seen in *Entamoeba histolytica* and its relationship with host microbiome. Among asymptomatic and amoebic liver abscess (ALA) patients, considerable alterations in 11 major microbiome populations were observed. The populations of *Bacteroides* spp., *Bifidobacterium* spp., *Lactobacillus* spp., and Clostridium spp. were especially significantly decreased. Similarly, asymptomatic patients showed a considerable decrease in commensal *E. coli* and an increase in Pseudomonas aeruginosa populations [29]. Interestingly, one of the characteristics that sets *Entamoeba* spp. apart from other protozoans (to a higher degree) is its nature of intestinal mucosal colonization. Protozoan, probably due to its ancient relationship with human hosts, has been reported to live in a commensal relationship with asymptomatic hosts, with non-pathogenic trophozoites releasing cysts on a continuous basis [30,31]. However, hamster model studies have suggested that invasiveness is triggered when a considerable amount of trophozoites feed on commensal bacteria, therefore inducing a release of enzymes, such as cysteine proteases. The resulting enzymatic activity triggers protozoan to become highly invasive in the intestinal environment, causing significant epithelial cell damage (observed as inflammations and lesions). Inflammation triggers the production of several cytokines, including interferon gamma (IFNγ), tumor necrosis factor (TNF), and interleukins (IL) 4, 5, 8, and 17. The following phagocytic activity causes dysbiosis (severe alteration of microbiota, disrupting the host–microbiome relationship). This has not only been observed to cause FGID symptoms, but it has also been observed to increase epithelial and endothelial permeability. This event aids *Entamoeba* spp. in infecting liver tissues through portal circulation, causing cell damage, lesions, and hepatic abscesses [30,32,33].

## 4. Cross-Organ Impacts

### 4.1. Gut–Liver Axis

Portal circulation results in the passage of *Entamoeba* spp. Through to the bloodstream, primarily reaching the liver. This event induces a host immune response, creating severe conditions, such as a high influx of cytotoxins, phagocytic nitrogen intermediates, and reactive oxygen species [34]. A noteworthy work on this interaction and the establishment of *E. histolytica* has been reported by Rigothier, et al. [35]. The study involved a hamster model of *E. histolytica* infection tracking through ^35^S-labelled protein monitoring. The study, continuing from previous models [36,37], indicated that trophozoites arriving through portal circulation enter the liver through sinusoids, causing host neutrophil reaction and resulting in localized inflammation. The host liver hepatocytes degenerate and lyse due to both neutrophil activity and trophozoite cytolytic activities. During this early phase (<12 h), a severe environment created by neutrophil activities cause massive trophozoite mortality. However, after 12 h, the trophozoites enter a commitment phase where considerable multiplication is observed. This causes the creation of numerous infection loci on liver parenchyma, ultimately causing numerous necrotic and inflammatory regions [35].

Although *Cryptosporidium* spp. and *Giardia* spp. are unable to permeate into the bloodstream through the intestinal epithelium, infection with these parasites impacts numerous organs beyond the gut. One of the early cryptosporidiosis mouse models showed that in an immunodeficient host, such as athymic and T-cell-depleted mice, the infection caused a swollen liver due to inflammation of the hepatic biliary system, possibly causing jaundice-like effects. Similarly, distension was seen across the bile and cystic ducts, in addition to the gall bladder [38]. Our mouse study indicated spiked oxalate levels in the hepatocytes during cryptosporidiosis, likely causing hyperoxaluria or a hyperoxaluria-like condition [20]. Similar cases have been observed in the case of *Giardia* infections. The alterations of microbiota caused during *Giardiasis* have been shown to cause nutrient malabsorption [39]. The observations from *Giardia*-infected children showed vitamin A malabsorption, not only from intestinal parasitism, but also indirectly via liver-based retinol mobilization. Our study [28] showed that depletion of *Akkermansiaceae* spp. in the gut caused oxidative stress across the gut–liver axis, leading to elevated glutathione metabolism, especially in the small intestine, serum, and liver.

### 4.2. Gut–Lung Axis

Although very uncommon, infection of the respiratory system by protozoan parasites has been reported. *Cryptosporidium* and *Entamoeba* have been known to cause opportunistic invasion of the lungs and bronchi [40]. Case-study subjects infected by *E. histolytica* displayed productive coughing, breathing issues, chest pain, and erythema as major symptoms. This was caused by dry oropharynx mucosal membranes, displaced right lung (especially, right and middle lobes) and right-sided pneumonia. combined with multicystic empyema [41]. Examination of respiratory fluids showed the presence of *Entamoeba* cysts and trophozoites. Another case displayed multi-organ infection, with trophozoites observed in the intestine and cerebrospinal fluid. However, symptoms such as dehydration and bronchiolitis also indicated a likelihood of *Entamoeba* infection in the respiratory system [42]. Similarly, multi-organ *Cryptosporidium* infection case studies have been presented. In one study, *C. parvum* was identified by 18S rDNA analysis of sputum (tracheal/bronchial mucus expelled during coughing) and stool samples of two patients, resulting in death [43]. The case studies performed by López-Vélez, et al. [44] and Clavel, et al. [45] involved diagnostics from sputum and broncho-alveolar lavage samples and indicated lung cryptosporidiosis in about 16% and 100%, respectively, of infected patients.

Enteric infection’s triggering of the host immune system of adolescents has been shown to cause hyper-activation of IFN and IL proteins, resulting in the release of cytokines, such as IL-1β and IL-10. Cytokine overexpression has previously been shown to cause obstructive issues in the respiratory system, such as inflammation and asthma [46]. Resultant α-1-antitrypsin release in stool samples [47] has already been used as enteropathy marker. Burgess, et al. [46] coupled this effect with wheezing monitoring as an earlier enteric infection biomarker system in 0–2-year-old infants. The study indicated wheezing episodes in 43% of the 700 infants observed. Higher wheezing incidence indicated increased IL-10 and IL-β levels, especially during early infancy (by the age of 24 weeks), whereas a successive increase in stool α-1-antitrypsin release (by the age of 40 weeks) resulted in decreased wheezing episodes. In our studies of both cryptosporidiosis [20] and *Giardiasis* [28], elevated accumulation of short-chain fatty acids (SCFAs) have been seen in the gut during infection. Gut–lung axis studies have shown that SCFAs are transferred from the gut to the lungs via serum, and they modulate the immune system [48] through the impairment of dendritic cells (DC). This causes an attenuated allergic response [49,50] and promotes regulatory T-cell differentiation, reducing asthmatic response [51] and reducing neutrophil recruitment during influenza [52].

Walker, et al. [53] indicated a direct relationship between occurrences of diarrhea and subsequent pneumonia (or pneumonia-like instances), with a relative risk of up to 1.08/day among Ghanaian children. Similarly, relative risk factors, such as zinc deficiency, increased the relative risk of mortality by 1.2 in both cases. A correlation between these two, as well as a zinc deficiency factor, can therefore be used, in combination with other factors, as a potential biomarkers for early diagnosis of protozoan infection. The review work by Halliez and Buret [54] discussed numerous long-term consequences of *Giardiasis* in extra-intestinal aspects. Some of the primary issues observed been caused by the malnutrition, immune response, and solute/mineral losses during diarrhea [55].

## 5. Nutritional Interventions

### 5.1. Prebiotics, Probiotics, and Synbiotics Supplementation

The phenomenon of FGIDs is not uncommon among post-infection patients. The long-term impacts of cryptosporidiosis and *Giardiasis*, including IBS, cognitive deficiencies, chronic fatigue syndrome, and joint pains, have been highlighted [54,56], although further studies are needed to establish the individual associations. The Rome Foundation, through its Rome IV working report, has indicated the significance of post-infection IBS (PI-IBS) [57]. The report indicates a 4–36% occurrence of PI-IBS among enteritis patients and highlights the complete absence of a pharmacologic strategy to treat PI-IBS.

In this context, probiotics have been investigated to address this problem. It has been suggested that in cases of post-infection diarrhea caused by parasites, such as Campylobacter, Salmonella, *Cryptosporidium*, and *Giardia*, the supplemented probiotics compete with the gut parasites for nutrition and resources, thereby increasing anti-parasitic immunity within the gut [58]. This in turn aided the shortening of the diarrheal period and reducing its severity. The predominant species used on commercial levels include *Lactobacillus*, *Streptococcus*, *Enterococcus*, *Lactococcus*, *Pediococcus*, *Bacillus*, *Escherichia*, and sometimes *Saccharomyces* [59]. The American Gastroenterology Association (AGA), in its 2020 guidelines, despite indicating that there are knowledge gaps regarding the impact of probiotics on post-infection conditions, conditionally recommended the use of probiotics during antibiotic treatment. Although a reduction in the diarrheal period by 21.91–28.9 h was seen in pediatric acute infectious gastroenteritis, the experimental outcomes varied, resulting in the AGA recommending against probiotic use during infectious gastroenteritis [60]. It is likely that this may be due to the symptom control and not targeting pathogenic mechanisms, former of which arguably remains the approach of the current treatments [61]. One of the recent studies exploring IBS caused from cow milk allergy may shed some light onto this change in approach. The study, based on the brain–gut immunoendocrine microbiota axis, indicated that the use of extensively hydrolyzed casein formula, along with the *Lactobacillus rhamnosus* GG probiotic, helped to decreased FGIDs in children [62]. Similarly, the intake of low fermentable oligosaccharides, disaccharides, monosaccharides, and polyols (FODMAPs) has been shown to help alleviate IBS [63]. This is not surprising since these molecules are not digestible by the human gut and act as good fiber sources.

Synbiotics and combination therapies have also been indicated to have positive impacts on gut health. Complementary synbiotics are the mixture of prebiotic and probiotic components. In an inflammatory bowel disease mouse model, Shinde, et al. [64] observed that synbiotic additives, created by mixing prebiotics (such as green-banana-resistant starch) and probiotics (such as Bacillus coagulans) led to an increase in colonic SCFAs. This phenomenon was not observed with the addition of probiotic-only supplementation, but was observed when prebiotics were added, indicating the important role of prebiotics in the production of SCFAs. In addition to gut modulation, synbiotic feeding has also been shown to increase the levels of anti-inflammatory and chemopreventive metabolites, such as 2-pentanone [65], which have been shown to inhibit prostaglandin and COX-2 protein expressions in colon cancer cells [66]. Furthermore, synbiotic therapy has been shown to alleviate chemotherapy effects, as recently reviewed by Singh, et al. [67].

In contrast, synergistic synbiotics consist of adding a stimulated microbe or enhancing the activity of a delivered microbe by adding a specific substrate. A good example can be cited through the work of Boger, et al. [68], who utilized short-chain inulin (sc-inulin) as a prebiotic and *Lactobacillus paracasei* subsp. *paracasei* W20 as a step 1 probiotic. The species was able to ferment sc-inulin which, through a cross-feeding mechanism, enabled increased fermentation by a step 2 probiotic, *Lactobacillus salivarius* W57. Such synergistic synbiotics, although much more difficult to obtain, have the ability to deliver significant impacts in a selective, targeted manner [69] such as to counter AMR and MDR. In the case of protozoal infection, an early study on rats by Ribeiro, et al. [70] presented supplementation of a synbiotic mixture of *Bifidobacterium* animalis and Raftilose^®^ P95 fructooligosaccharides on its own and as an addition to dexamethasone treatment.

Metronidazole (MTZ) is the most used drug to address *Giardiasis*. However, emerging studies are showing an increasingly developing *Giardia* resistance to MTZ [71,72]. In this context, nutritional sources have shown promise in mitigating parasite removal. For example, blueberry polyphenolic extract, under in vitro conditions, has been shown to inactivate > 90% *Giardia* trophozoites at 167 µg/mL, with respect to 100% achieved by 67 µg/mL MTZ [73]. The dichloromethane polyphenolic extracts of ginger and cinnamon, particularly at a 20 mg dosage rate in albino rats, have been shown to reduce *Giardia* cyst count by 90.1% and 100%, respectively, while reducing cyst count by 75.4% and 34.1%, respectively [74]. In a Swiss mouse model, BIOintestil^®^ (which contains gingergrass (or palmarosa) essential oil, ginger powder, and gingerol), when combined with MicrobiomeX^®^ (which contains citrus extract flavonoids) at a 100 mg/day dosage, eliminated 100% of *Giardia* cysts within five days and was twice as effective as albendazole and metronidazole [75].

Similarly, the mouse model study indicated that zinc supplementation and *Lactobacillus* acidophilus + dill seed oil supplementation reduced *Cryptosporidium* oocysts by 98.3% and 95.8%, respectively, with respect to the prescribed drug, nitazoxanide (91.6%), within eight days of treatment. Furthermore, these treatments were also found to significantly reduce TNF-α levels in serum [76].

Although supplementation of micronutrients such as zinc have been suggested to control parasitic activity in gut [72], it has been shown that in the infant gut, supplementation of micronutrients such as iron and vitamins (A, C, D, folate), but without zinc, decrease gut microbiome diversity and aid the growth of protozoal parasites, such as *Entamoeba* [77]. Therefore, it is important to ascertain the impact of certain nutritional interventions.

### 5.2. Postbiotics and Microbiome Modulation to Improve MDR Resilience

Postbiotics, although traditionally neglected in modern medicine for FGID treatments, have started to gain relevance over the last few years. The International Scientific Association for Probiotics and Prebiotics defines postbiotics as the “preparation of inanimate microorganisms and/or their components that confers a health benefit on the host” [78]. They can range from sugar alcohols to amino acids, fatty acids, vitamins, and microbial peptides. Some notable examples include vitamin K generated by *Escherichia coli* in gut, SCFAs, D-amino acids, and small proteins/peptides.

SCFAs and D-amino acids, generated by gut microbes, have been shown to contribute towards the host immune response to infections [64,79,80,81]. In our mouse model study of cryptosporidiosis, SCFA accumulation elevated in response to the infection. Particularly, significant acetate elevations were seen in the duodenum and jejunum, while butyrate levels increased in the caecum and colon [20]. However, it appeared that these levels were much higher during *Giardiasis* infection, particularly the propanoate and butyrate increase in the colon [28]. It has been shown that the supplementation of SCFAs, such as acetate, to influenza-infected mice through the gut–lung axis aided the improvement in alveolar macrophage activity [82]. On the other hand, propionate production by gut *Bacteroides* disrupted intracellular homeostasis, inhibiting pathogenic *Salmonella enterica* growth [83]. In the aging mice, *L. acidophilus* DDS-1, when added as a probiotic, led to an increase in caecal butyrate levels, leading to the downregulation of inflammatory cytokines [84]. Similarly, D-amino acid levels increased throughout the mouse gut during cryptosporidiosis [20] and *Giardiasis* [28] infection, but their levels were observed to be much higher in the small intestine with respect to the large intestine. Early assessment has shown that although D-amino acids were unable to prevent *Staphylococcus aureus* colonization, they inhibited biofilm assembly development under in vitro conditions [85].

In addition to SCFAs and D-amino acids, small proteins such as bacteriocins/colicins produced by gut Enterobacteriaceae have been shown to competitively inhibit the growth of pathogenic *Salmonella enterica* [86]. One of the very recent reviews by Upatissa and Mitchell [87] indicated the utilization of these small proteins to control specific drug-resistant pathogens. For example, microcin J25 has been shown to inhibit more than 28 multi-antibiotic-resistant *Salmonella enterica* serovars [88]. The protein also has shown effectiveness against some strains of multi-drug-resistant *E. coli* [89]. The work of Upatissa and Mitchell [87] provides a good insight into these proteins and their action mechanisms. In the case of cryptosporidiosis, cathelicidin-related anti-microbial peptides (CRAMPs) have been indicated to significantly reduce the parasite burden. However, the indigenous CRAMP appeared to be downregulated during cryptosporidiosis. In such a case, oral feeding of 5 µg CRAMP has been shown to aid the reduction in *Cryptosporidium* sporozoites, but not that of the oocyst [90]. Anti-microbial peptides, particularly from venomous insects such as bees, have shown inhibitory effects on protozoal parasites [91,92] and promise to be applied in protozoal infection treatment.

One of the components that may arguably be categorized as both prebiotics and postbiotics are enteric viruses, especially phages. In addition to chronic disorders such as colitis [93], they have been proposed as an effective treatment for AMR and MDR pathogens [94,95,96]. For example, phages such as Bϕ-B1251 and PD-6A3 have been shown to provide lytic activity against MDR-resistant Acinetobacter baumannii [97], *E. coli*, and methicillin-resistant *Staphylococcus aureus* [98], respectively. Furthermore, most recent studies have shown the use of CRISPR-Cas-carrying bacteriophages (also called CR-Phages) to remove MDR bacterial species under in vitro [99] and mouse model conditions [100]. However, due to their extreme specificity, more studies need to be undertaken to ascertain the impacts of bacteriophage treatment [101].

In addition, recent reviews of Strati, et al. [102] have also covered various new and emerging techniques to improve gut microbiome resilience and microbiome resurrection post-infection and in various other gut and extra-gut inflammations.

## 6. Application of Multiomics in High-Throughput Analysis of Gut Microbiome Health and Inter-Organ Axes

### 6.1. Multiomics Approaches

Numerous analytical techniques have been used to elaborate the workings of this multi-level complex relationship. Although the technical specifications of individual omics platforms are beyond the scope of this work, mentioning these platforms is important in the context of analytical assessment of the abovementioned systems biology approaches. Pinu, et al. [103] provided excellent coverage of multiomics integration, including experimental design, data integration and analysis, systems modelling, and challenges.

In metagenomics, emerging sequencing methods, combined with robust databases [104], provide a good understanding of microbiome identification and characterization. The compiled work of Nagarajan [105] provided further information. Metaproteomics have also been significantly developed over last few years, with their own databases and highly sensitive, high-throughput mass-spectrometry-based analytical tools. Detailed information of these instruments, their relevant sample preparation, work-flows, data quantification, and synchronization have been reviewed by Antoine and Bruno [106] and Heyer, et al. [107]. Metabolomics, including lipidomics, glycomics, and ionomics, have the potential to provide biochemical information to understand and characterize various mechanisms. Metabolomics have been applied to investigate bacterial processes related to preventative health [108,109], environmental pollution [110,111], and food [112], among others. Metabolomics also assist in extricating the correlation between cell phenotypes and their metabolic patterns and stoichiometry [113]. Metabolic flux studies have previously been applied in toxicology and medicine [114,115,116], as well as microbial respiratory systems [117,118]. For further information, readers are recommended to refer to the works of Beale, et al. [119].

In the case of enteric infection, a highly complex host–parasite–host–microbiome relationship is developed. Resultantly, a significant change in genomic, transcriptomic, proteomic, and metabolic expression is observed, contributing to the expressional pool. This variation causes considerable perturbations at various regulation levels, such as gene expression (major) and proteomic translation (considerable, but less than gene expression) (Figure 4).

Multiomics utilizes the capability of two or more omic platforms, either in a standalone or integrated manner. Since the individual omic platforms do not provide the big picture of systems biology, an integrated approach is now being increasingly utilized. In the context of gut infections, our recent works have integrated metabolomics with gut metaproteomics and 16S rRNA genomics to understand the interactomics of cryptosporidiosis [20] and *Giardiasis* [28]. Particularly in the *Giardiasis* interactomics study [28], we integrated the 16S rRNA population genetics-GC-MS metabolomics data with LC-MS metaproteomics-GC-MS metabolomics via the PICRUSt [121], BURRITO [122], and MetaboAnalyst [123] toolboxes. The integration not only indicated the key microbiome species impacted, but it also helped to filter the most significant redox pathways impacting the gut–liver axis. Similar strategies have been further employed to understand the extra-gut mechanism of FGIDs, such as IBS, functional dyspepsia [124], and infections such as SARS-CoV2 [125,126]. In the area of the nutritional impacts of prebiotics and probiotics, the study reported by Shinde, et al. [64] included the combined approach of immunohistochemistry, enzyme kinetics, and metabolomics, including SCFA analytics. The study showed that probiotic and prebiotic combinations provide synergistic immune-regulating efficacy, protect epithelial integrity, and mediate the reduction in colonic inflammation. The 16S rRNA population genetics data were added to this approach to determine the modulation of key gut microbial species during these treatment regimens to improve gut health in a mouse model [79,84]. The on-field study conducted by Attia, et al. [127] utilized a less sensitive but more impactful multiomics approach to determine the impacts of severe acute malnutrition (SAM) contributing to mortality in pediatric patients (age: 6–60 months). The study undertook targeted genomic identification, protein, and immune assays, combined with GC-MS-based SCFA analysis. The results indicated a considerable presence of Shigella, *Giardia*, and Campylobacter, with upregulated calprotectin and depleted butyrate propionate in fatal cases. A more recent omics study [128] indicated that in the case of SAM, the pre-treated pediatric patients had a higher number of Proteobacteria, particularly Enterobacteriaceae. Furthermore, they had lower gut microbiome diversity and depleted SCFAs. When fed with high cowpea flour, combined with WHO standard feed F75 and F100, the impacts of the antibiotics decreased, and gut integrity improved. The study indicated that post-antibiotic cessation, children fed with this diet showed increased *Firmicutes*, correlating to increased SCFA by the 28th day after intervention.

### 6.2. Application of Artificial Intelligence and Machine Learning (AIML) and Future Aspects

AIML has been making strong inroads in systems biology. In addition to multivariate statistics, it provides strong potential for understanding the key biomarkers or pathways involved in gut processes. A recent study by Muller Bark, et al. [129] applied machine learning (ML) to elaborate the metabolome and lipidome associations of glioblastoma patients through graphical network analysis. The ML analysis indicated homogenous networks with lipid cluster linkages between the key metabolic markers. On the other hand, among the patients with unfavorable outcomes, fewer cluster networks were seen, with altered key lipids. Another study assessed host–pathogen interaction and its effects on the gut microbiota through machine learning [130]. The study involved a minimum curvilinear Markov clustering (MC-MCL) method to analyze mechanisms of bacterial network re-organizations caused by proton pump inhibitor (PPI) intervention and Helicobacter pylori infection in the stomach. MC-MCL indicated that nine bacterial species from Fusobacteria, Proteobacteria, *Bacteroides*, and *Firmicutes* were positively correlated with the PPI treatment. Similarly, six species showed depletion during H. pylori infection. Another approach for diagnosing several neglected tropical diseases was reported by Arnold, et al. [131]. The study used an ensemble algorithm to cross-validate antibody response from variable populations to predict pathogenic surveillance. These approaches, as reported previously by the CoviRx database for COVID-19 [132] in addition to multiomics, has the potential to provide much more rapid and reliable outcomes of nutritional solutions to protozoal MDR infection agents.

The importance of AIML in multiomics studies increases even further when the question of “which nutritional combinations not to add?” is asked, in order to modulate MDR or AMR. For example, monensins are generally used in dairy calves as coccisoid control agents and growth promoters. However, a very recent study [133] indicated that diets containing monensin increased the risk of multi-drug-resistant *E. coli* when compared to an essential-oil-containing (garlic extract + carvacrol, capsaicin, cinnamaldehyde blend) diet. The AIML platform, when combined with multiomics at very early stages of nutritional formulation, can prevent or minimize these issues at more advanced stages.

Recent reviews [134,135,136] have highlighted the impact of integrating multiomics with AIML to further develop the prebiotic, probiotic, and symbiotic landscape, which the readers can refer to for more information.

Multiomics, combined with AIML, have started to show promising results in understanding the mechanisms of these interventions and their impacts. While multiomics has shown immense potential to obtain more understanding of infection interactomics and the impacts of nutritional interventions, AIML still remains in its infancy. A combination of portable multiomics equipment, cloud computing, and AIML would expedite our capabilities to address the research and development gaps in this area over the upcoming decade.

## Figures and Tables

**Figure 2 microorganisms-11-01800-f002:**
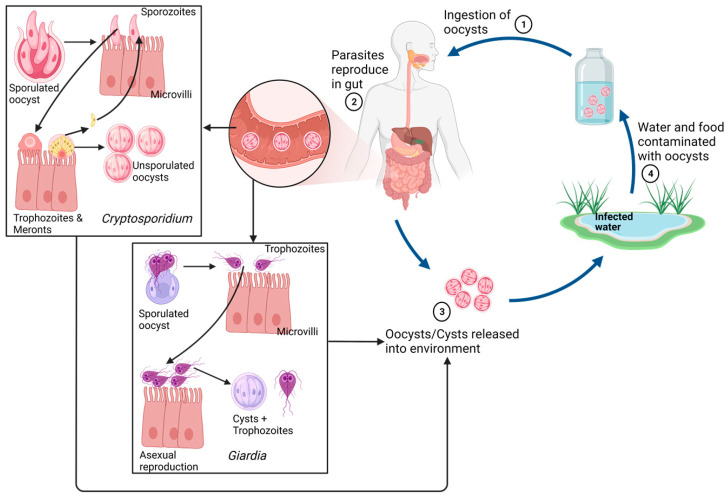
A general life cycle and growth stages of protozoan enteric parasites, with *Cryptosporidium* and *Giardia* as examples. **Note**: This is a generalized outlay, with each protozoan discussed here showing minor variations in their respective cycles. Other protozoal parasites show slightly different life cycles than the ones represented. Created with Biorender.com.

**Figure 3 microorganisms-11-01800-f003:**
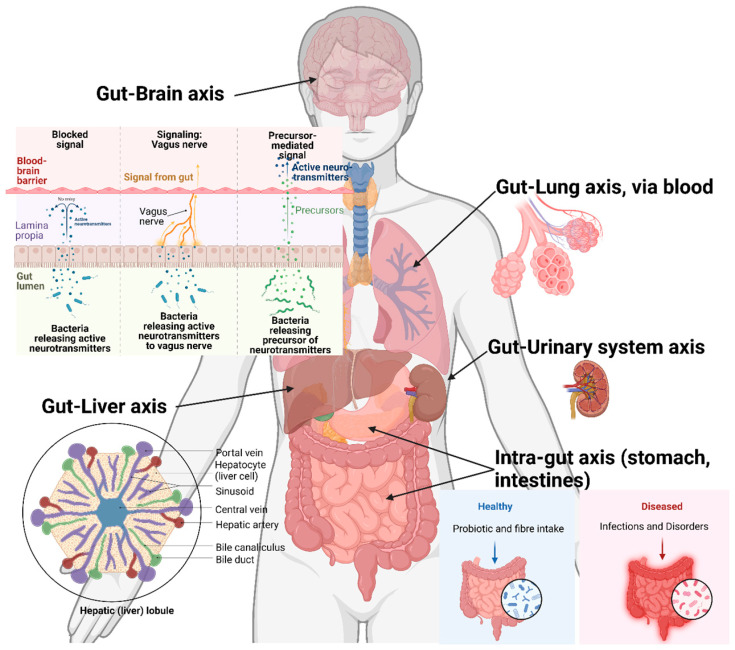
A general representation of the host organs directly and indirectly affected by the enteric protozoan parasite infection. Created with Biorender.com.

**Figure 4 microorganisms-11-01800-f004:**
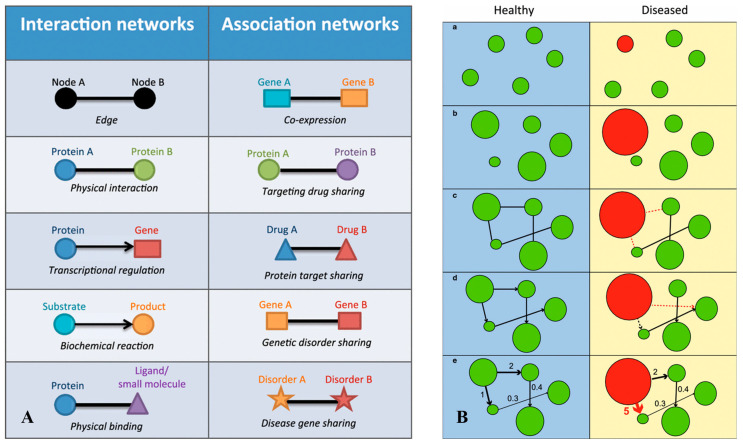
Interaction and association networks: (**A**) representing regulatory functions and functional expression at various levels and (**B**) representing the changes caused during disease, with regulatory components of (**a**) present/absent key component (green: presence, red: absence); (**b**) misregulated gene expression causing over/underexpression (node size: expression level); (**c**) absence/erroneous interactions (dotted lines represent erroneous interactions); (**d**) misregulated directions (misdirected arrows); (**e**) interaction impact (arrow thicknesses + numbers). Figure courtesy of Jinawath, et al. [120]. Distributed under the terms of the Creative Commons Attribution 4.0 International License (http://creativecommons.org/licenses/by/4.0/) (accessed on 30 April 2023).

## Data Availability

Not applicable.

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
