# Peer review of "Intelligent Biological Networks: Improving Anti-Microbial Resistance Resilience through Nutritional Interventions to Understand Protozoal Gut Infections"

_microorganisms, 2023, doi:10.3390/microorganisms11071800_

Round 1
Reviewer 1 Report
Abstract
Line 17: “The robustness of these pathogens means requirement of harsh treatment methods such as medicinal drugs or antibiotics. However, such treatments impact the gut microbiome, and create dysbiosis, often leading to opportunistic pathogens, anti-microbial resistance, or functional gastrointestinal disorders…” Not only the treatment with drugs, but also the protozoan infection itself can induce dysbiosis…
Line 31: “This manuscript is organised in sections..” Authors should include the last section that discuss analytical techniques to study interaction and association networks
Introduction/Current Global status
I think that these topics could be merged. Besides, authors could try to update the epidemiological data, as the Figure 1 presents the global distribution of deaths due to diarrhea in 2012.
GBD 2016 Diarrhoeal Disease CollaboratorsEstimates of the global, regional, and national morbidity, mortality, and aetiologies of diarrhoea in 195 countries: a systematic analysis for the Global Burden of Disease Study 2016, The Lancet Infectious Diseases, Volume 18, Issue 11, 2018, Pages 1211-1228, ISSN 1473-3099, https://doi.org/10.1016/S1473-3099(18)30362-1.
Lines 50-55: So far, the intestinal Entamoeba species considered pathogenic to humans is Entamoeba histolytica. Please write the species name all over the text and tables avoiding misinterpretation.
I suggest adding to the “Current Global Status” that outbreaks of intestinal waterborne protozoan are mostly observed in developed countries and is less common in developing/underdeveloped ones. Also, although related to childhood diarrhea, in endemic countries these infections are usually asymptomatic, and only diagnosed during coprological surveys of intestinal parasites in daycare or elementary school children.
Table 1:
- I suggest deleting agent characteristics (parasite dimensions). I think this information is not necessary. Also, there are two size ranges for the cysts of Entamoeba spp., the higher one is for Entamoeba coli, considered non-pathogenic;
- All infections, especially in developing countries, such Brazil, are mainly asymptomatic. Authors should include this at the “health symptoms”; and maybe change to “clinical presentation”;
- Giardia and Cryptosporidium can infect humans and domestic and wild animals;
- Entamoeba histolytica can infect humans e and nonhumans primates;
SUGGESTION: Alternatively, authors could delete both Figure 1 and Table 1. Considering the subject of this review, they can be removed from manuscript without any prejudice. Keep the brief introduction/global status with epidemiological data of parasites and then the transmission and life cycle topic with the corrections pointed bellow.
Lines 85-88: …” The mature cyst in Entamoeba, Cryptosporidium and Giardia consists of 4, 4 and 2 trophozoites (sporozoites in Giardia), respectively….
Sporozoites are parasitic stages observed in Apicomplexa protozoan, such as Cryptosporidium. I suggesting changing to: The infection occurs by the ingestion of the tetranucleated cysts (Entamoeba histolytica and Giardia) or sporulated oocysts (Cryptosporidium). Acidic conditions in the stomach, followed by slightly alkaline conditions in the upper intestine (duodenum) result in the process of excystation, releasing these trophozoites or sporozoites in the duodenum [10,11].
Lines 89-98: Because the peculiarities of each parasite, the paragraph is sometimes confuse. Please separate information maintaining it brief. See some examples of confusing sentences:
“But sometimes, particularly with Entamoeba, the trophozoites do penetrate the mucosal…” this mechanism of intestinal invasion is typical of Entamoeba histolytica (pathogenic cycle). Giardia trophozoites attach to the microvilli through their ventral disk, flagellar movements, and a variety of proteins.
“The Cryptosporidium trophozoites can opt?? for both sexual (gemetogeny) and asexual (merogeny) reproduction…”
Cryptosporidium (compulsorily) undergoes asexual and then sexual multiplication, generating the oocyst which sporulate and is eliminated in feces.
Lines 100-106: This paragraph can be rewritten for better understanding: all 3 parasites can cause asymptomatic infections with low parasitic load which delay diagnosis; small intestine is the habitat of Giardia and Cryptosporidium, Entamoeba histolytica lives at the large intestine; infections can be localized or spread to other organs depending (mainly) on the nutritional or immune status of hosts.
Figure 2: This cycle is completely wrong; authors mixed the life cycle of Cryptosporidium with Giardia. The multiplication and type of intestinal colonization are completely different; Cryptosporidium is an obligatory intracellular parasite. I also suggest changing the legend to “A general representation of host-organs directly and indirectly affected by enteric infections.” As this can happen due to other intestinal pathogen infections.
Dysbiosis section Line 132: Delete repeated word “increase” ….maturation, showing a 2-3 fold increase in population increase of Cryptosporidium under....
Line 133: “In a similar study with Coquerel’s sifaka, a Madagascar lemur species, genomic studies indicated a considerable microbial diversity depletion with recovery depending on the host age (older lemurs recovered earlier than younger ones).”
The previous study cited by authors (ref #16) showed that biofilms in an aquatic environment can support the multiplication of Cryptosporidium (historically, it has always been assumed that Cryptosporidium, like other apicomplexans, can only multiply intracellularly in host intestinal cells). So, how this second study (depletion of primate microbiome by Cryptosporidium infection) is similar to the previous one? Please, clarify.
Line 155-158: The term sporozoite is usually used for apicomplexans infecting stages, please change to trophozoites for Giardia and Entamoeba histolytica
Line 183-187: “… This induces a release of amoebaporic enzymes such as cysteine proteases. The resulting enzymatic activity…”
E. histolytica cysteine proteases cleave the mucin and sIgA antibodies. Amoebapores are a family of lytic peptides secreted by E. histolytica that are inserted into the membrane of enterocytes causing their lysis, similar to granulysin from cytotoxic cells. Please check the sentence above.
Line 214: “One of the early mouse models for Cryptosporidium spp. showed that the infection caused a swollen liver, due to inflammation of hepatic biliary system, possibly causing jaundice-like effects”
The mouse models of Ungar et al (ref #37) for chronic symptomatic cryptosporidiosis were adult athymic mice and T-cell depleted mice. Therefore, considering the effects of this parasite to the gut-liver axis, it is important to point out that dissemination of protozoan to extraintestinal organs occurs only in immunocompromised hosts (as was done at the topic gut-lung axis).
Lines 234-236: Change cysts for trophozoites, the parasite stage observed in hosts fluids.
Line 467: Change to italics scientific names
Author Response
Dear reviewer,
Thank you for the elaborate peer review. We have incorporated almost all the suggestions which were suggested. We are sure that these will definitely improve this manuscript.

Reviewer 2 Report
The review titled Intelligent Biological Networks: Improving anti-microbial resistance resilience through nutritional interventions to understand protozoal gut infections presents good knowledge to the medical community on enteric protozoa and their relation with antimicrobial resistance through nutrition. Some points need improvement.
1. Line 62-63 add a full stop after the Indian subcontinent.
2. Entamoeba also affects the animals add this to Table 1 in the raw of the Entamoeba host and add a reference in the table and list of references.
3. lines 85-86 correct this The mature cyst in Entamoeba, Cryptosporidium, and Giardia consists of 4, 4, and 2 trophozoites (sporozoites in Giardia) as Cryptosporidium has oocyst not cyst so indicates that separate from the cyst of Giardia and Entamoeba also the content of the oocyst is sporozoites while the others are containing nuclei, not sporozoites so please rewrite this sentence again.
4. Line 88 please correct releasing these trophozoites in the duodenum to sporozoites and trophozoites
5. Line 95 Giardia sporozoites, on the other hand, reproduce please correct not sporozoites trophozoites as you mentioned above.
6. Line 97 The Cryptosporidium trophozoites can opt please add in epithelial cells of the intestine. Because the reader may think reproduction is in the gut lumen.
7. In Figure 2 in the gut, the oocyst of cryptosporidium is sporulated not unsporulated so please correct.
8. Line 391 biotics are enteric viruses, especially macrophages please correct to the name of the virus either phage or other, not the macrophage the immune cell.
9. Line 398 In addition, recent reviews of [99] has also please add the name of the author of this paper in the citation.
10. Line 478 A recent study by [126] applied machine learning add the author's name to the citation.
11. Line 490 was reported by [128]. The study add the author's name to the citation.
12. Line 519 (accessed on 17 January). Indicate the year.
English needs proof reading of typing errors.
Author Response

(The authors gave the same response as above.)

Reviewer 3 Report
Really the introduction is very brief and is not really an introduction.
The objectives of the manuscript must be clearly defined and presented.
The methodology employed to acquire, select and decide to use references must be presented in brief (even this not being a structured review).
Sections 3 and 4 can be merged and shortened as they are not the key parts of the manuscript.
I suggest to add a new section 7 to present an opinionated account of the points presented in 5 and 6.
Please add a concluding section.
Overall: revision and re-evaluation.
Moderate editing of English language required.
Please avoid Australian colloquialisms, as these are not understood worldwide.
Author Response

(The authors gave the same response as above.)

Round 2
Reviewer 3 Report
The manuscript has been improved.
A final point before acceptance: please carry out a literature search to make sure no relevant references have been published during the last weeks after submission of the manuscript, in order to assess if they merit inclusion in the final paper.
Author Response
We thank the reviewer for this suggestion. We have added the new publications which have come between April 2023 and June 2023.
